# Centronuclear Myopathy Caused by Defective Membrane Remodelling of Dynamin 2 and BIN1 Variants

**DOI:** 10.3390/ijms23116274

**Published:** 2022-06-03

**Authors:** Kenshiro Fujise, Satoru Noguchi, Tetsuya Takeda

**Affiliations:** 1Departments of Neuroscience and Cell Biology, Howard Hughes Medical Institute, Yale University School of Medicine, New Haven, CT 06520-8001, USA; kenshiro.fujise@yale.edu; 2National Institute of Neuroscience, National Center of Neurology and Psychiatry (NCNP), Tokyo 187-8502, Japan; noguchi@ncnp.go.jp; 3Department of Biochemistry, Faculty of Medicine, Dentistry and Pharmaceutical Sciences, Okayama University, Shikata-cho 2-5-1, Kita-ku, Okayama 700-8558, Japan

**Keywords:** centronuclear myopathy, T-tubules, dynamin, BIN1, membrane remodelling

## Abstract

Centronuclear myopathy (CNM) is a congenital myopathy characterised by centralised nuclei in skeletal myofibers. T-tubules, sarcolemmal invaginations required for excitation-contraction coupling, are disorganised in the skeletal muscles of CNM patients. Previous studies showed that various endocytic proteins are involved in T-tubule biogenesis and their dysfunction is tightly associated with CNM pathogenesis. *DNM2* and *BIN1* are two causative genes for CNM that encode essential membrane remodelling proteins in endocytosis, dynamin 2 and BIN1, respectively. In this review, we overview the functions of dynamin 2 and BIN1 in T-tubule biogenesis and discuss how their dysfunction in membrane remodelling leads to CNM pathogenesis.

## 1. Introduction

Centronuclear myopathy (CNM) is a hereditary muscular disorder that is diagnosed by the clinical features of congenital myopathy and pathological characteristic of centralised nuclei in the skeletal muscle biopsy [1]. Clinical features of CNM patients are broad spectra of onset age and symptoms, and the disease course of an individual patient is often unpredictable. Seven causative genes for CNM, *MTM1*, *SPEG*, *BIN1, DNM*2, *RYR1*, *TTN* and *CCDC78*, have been identified [2,3,4,5]. Among these CNM causative genes, *MTM1*, *BIN1* or *DNM2* variants cause disorganisation of T-tubules (transverse tubules) and triads in the skeletal muscle, suggesting their function in a common pathway during T-tubule biogenesis (Figure 1) [6]. *DNM2* and *BIN1* encode essential membrane remodelling proteins, dynamin 2 and BIN1 (also called amphiphysin II), respectively, and they are required for T-tubule biogenesis in skeletal muscle development [6,7]. *BIN1*- and *DNM2*-associated CNM patients show normal or slightly elevated levels of serum creatine kinase and slowly progressive muscle weakness [8,9,10,11]. This review will overview the functions of dynamin 2 and BIN1 in T-tubule biogenesis and discuss possible pathogenic mechanisms of CNM caused by their membrane remodelling defects, aiming for compensating other excellent reviews [2,12,13,14,15,16].

## 2. T-Tubules: Sarcolemmal Invaginations Essential for E-C Coupling

Rapid and coordinated contraction of striated muscles is achieved by coupled voltage- and calcium-dependent processes called excitation-contraction (E-C) coupling [17]. T-tubules are sarcolemmal invaginations required for the E-C coupling in both skeletal- and cardiac muscles [6]. In skeletal muscle cells, T-tubules are associated with sarcoplasmic reticulum (SR) on either side to form closely apposed membrane contacts termed “triad”, whereas, in cardiac muscle cells, their contact occurs only on one side to form “diad”. T-tubules are enriched with specific lipids such as PI(4,5)P_2_ (phosphatidylinositol 4,5-bisphosphate) and cholesterol and they serve as a platform for localizing T-tubule specific ion channels or lipid-binding proteins [18,19,20,21]. In the E-C coupling, excitation (depolarisation) of the T-tubule membrane induces conformational changes of a voltage-gated L-type calcium channel DHPR (dihydropyridine receptors), which in turn opens RyR1 (ryanodine receptor 1), a Ca^2+^ channel on SR, to allow Ca^2+^ release from SR to induce muscle contraction [22]. In the skeletal muscle cells, DHPR directly interacts with RyR1 to enable rapid signal transmission (within 2 ms) [23,24,25]. Thus, the structural and functional integrity of T-tubules is crucial for proper E-C coupling of the skeletal muscles. Not surprisingly, abnormalities in T-tubule structures cause various muscle diseases including congenital myopathies [26].

## 3. BIN1: A BAR Domain Protein-Inducing Membrane Curvature

### 3.1. BIN1 Functions in T-Tubule Biogenesis

BIN1 (Bridging Integrator 1) belongs to the conserved BAR domain superfamily that senses and induces membrane curvature [27,28,29]. BIN1 contains an N-terminal amphipathic helix Bin/Amphiphysin/Rvs-homology (N-BAR) domain that forms a “crescent-shaped” dimer, and its positively charged concave surface binds to negatively charged phospholipids to induce membrane curvature [30]. Human and mouse BIN1 are alternatively spliced to express tissue-specific isoforms [27,31,32,33]. The skeletal muscle-specific BIN1 isoform, isoform 8, has been shown to localise on T-tubules [33]. Conditional *Bin1* knockout mice in skeletal muscle exhibit neonatal lethality [34] and acute knock-down of BIN1 in adult mice caused structural and functional defects of T-tubules [35], indicating that BIN1 plays essential roles in the development and maintenance of the skeletal muscle. The BIN1 ortholog in *Drosophila*, Amph, is also required for muscle contraction, but not for synaptic vesicle trafficking, suggesting that it has a similar function as human BIN1 [36].

BIN1 isoform 8 consists of four functional domains: H0, N-BAR, PI and Src homology 3 (SH3) domains from N- to C-terminus [27] (Figure 2). H0 is an amphipathic helix that is folded and inserted into one leaflet of the membrane to initiate oligomerisation of N-BAR domains and membrane tubulation [37,38]. N-BAR domain of BIN1 induces clustering of PI(4,5)P_2_ and in turn, recruits a downstream partner dynamin 2 to enhance membrane tubulation in T-tubule biogenesis [39,40,41]. Thus, BIN1 contributes to T-tubule biogenesis by regulating lipid composition and protein interaction in a positive feedback manner. N-BAR domain of BIN1 also interacts with F-actin to regulate its organisation via stabilisation or bundle formation of actin filaments [42]. Actin regulatory function of BIN1 is required for proper T-tubule biogenesis in cardiac muscle cells [43]. In contrast, the formation of BIN1-mediated T-tubule like structures (TLS) in mouse myoblast C2C12 cells is antagonised by actin polymerisation [44]. The PI domain that exists only in BIN1 isoform 8 interacts with PI(4,5)P_2_ [44]. Neuronal BIN1 isoform 1 that lacks the PI domain diffusely localises in the cytoplasm of CHO cells, suggesting essential roles of the PI domain in membrane invaginations required for T-tubule biogenesis [19]. Indeed, skipping of the PI domain in BIN1 by dysregulated alternative splicing causes aberrant T-tubule formation in CNM and myotonic dystrophy [45,46]. Lack of the PI domain does not affect muscle development *per se*, but it causes defects in the formation of T-tubule network and muscle regeneration due to a reduced pool of satellite cells [33]. The C-terminal SH3 domain of BIN1 interacts with PR domain-containing proteins such as dynamin 2 [9,47]. The SH3 domain of BIN1 also binds to its PI domain intramolecularly to form a closed auto-inhibitory conformation [41]. The autoinhibition of BIN1 is released upon PI(4,5)P_2_ binding to the PI domain that in turn recruits its partner proteins dynamin 2 and myotubularin to the PI(4,5)P_2_-rich membrane domains [41,48,49]. Interestingly, deletion of exon 20 that encodes the SH3 domain of BIN1 causes defects in T-tubule formation at E18.5 embryonic muscle fibres, but the triad structures in adult skeletal muscle are not affected [33]. This result suggests that the BIN1 SH3 domain is required for T-tubule formation, but not for its maintenance, at the early stages of skeletal muscle development.

### 3.2. CNM Pathogenesis Caused by Defective Membrane Remodelling of BIN1 Variants

Multiple pathogenic BIN1 variants have been identified in CNM patients (Figure 2). CNM-associated variants in the H0 helix, K21del, R24C and K35N, have been reported to cause abnormalities in T-tubule structures due to decreased abilities to generate membrane curvature [44,48]. CNM-associated variants in the N-BAR domain, D151N and R154Q, are defective both in membrane binding and in curvature sensing possibly due to oligomerisation defects [33,44]. D151N is also defective in the clustering of PI(4,5)P_2_ both *in cellulo* and in vitro systems using a flat membrane sheet [39]. Another variant in the N-BAR domain, D149N, exhibits decreased membrane deformation abilities *in cellulo* [40]. Since membrane tubulation defects of K35N and D149N can be restored by supplementing with PI(4,5)P_2_, these variants are deficient in recruiting PI(4,5)P_2_ probably due to reduced membrane binding affinity [40]. CNM-associated variant IVS10-1G>A in exon 11 causes deletion of the PI domain, resulting in defective triad formation both in humans and dogs [45]. Thus, CNM-associated variants in H0, N-BAR and PI domains are likely to induce abnormal T-tubule structures due to their membrane deformation disabilities.

Two recessive CNM variants of BIN1, Q434X and K436X, that partially truncate the SH3 domain shows suppressed interaction with dynamin 2 [9,41,47]. In the skeletal muscle biopsies from CNM patients with these variants, abnormal T-tubule morphology with aggregated caveolae-positive membranous structures is observed [49]. Partial truncation of the SH3 domain by Q434X and K436X variants also keeps BIN1 in a constitutively open conformation with altered membrane deformation abilities [41]. The loss of autoinhibition by the CNM mutant BIN1 also causes enhanced interaction with myotubularin, which is a phosphatidylinositol-3-phosphatase for PI(3)P or PI(3,5)P_2_ encoded by a CNM causative gene *MTM1* [50]. The SH3 domain of BIN1 also interacts with N-WASP, an activator of Arp2/3 dependent actin polymerisation [51]. BIN1 mutants with truncated SH3 show suppressed N-WASP interaction and induce collapsed T-tubule structures [51]. Thus, the structural abnormalities of T-tubules caused by CNM-associated BIN1 variants are caused by abnormal protein–protein and/or protein–lipid interactions.

## 4. Dynamin: A Membrane Fission Catalyser in Endocytosis

### 4.1. Structure and Function of Dynamin

Dynamin is a large GTPase essential for membrane fission in clathrin-dependent and independent endocytic pathways [52,53,54]. There are three dynamin isoforms in mammals: dynamin 1 and 3, two tissue-specific isoforms highly expressed in neurons, and dynamin 2, a ubiquitously expressed isoform [55,56,57]. These isoforms are similar in amino acid sequences and share the same functional domains: G, middle, pleckstrin homology (PH), GTPase effector (GE) and PR domains from N- to C-terminus (Figure 2). The G domain is responsible for GTP binding and hydrolysis [58]. The middle and GED form a “stalk” structure that serves as interacting platforms in the formation of dimer or tetramer [59]. PH domain binds to negatively charged phospholipids such as PI(4,5)P_2_ and plays a role in clustering the phosphoinositides [60,61]. PH domain also senses membrane curvature by being hydrophobically inserted into the lipid bilayer [62]. Furthermore, the PH domain can bind to stalk structure intramolecularly to form autoinhibitory “closed” conformation that prevents untimely self-assembly [63]. The C-terminal PR domain binds to other SH3 domain-containing proteins such as BIN1, amphiphysin 1, and endophilin [9,64,65]. PR domain is also involved in actin organisation at invadosomes, membranous protrusions required for myoblast fusion [64,66].

Structural studies using cryo-EM, X-ray crystallography and high-speed atomic force microscopy (HS-AFM) gave mechanistic insights into dynamin-mediated membrane fission. Dynamin exists as a tetramer in a physiological condition in the absence of lipids [63], while it assembles into a helical polymer at the neck of endocytic pits [65] or on membrane tubules reconstituted in vitro from liposomes [67,68]. Conformational changes of dynamin helical polymer coupled with binding and hydrolysis of GTP promote membrane constriction and fission [69,70]. Although precise mechanisms of the dynamin-mediated membrane fission are still under debate, a few decades of studies in the past strongly support the following consensus views: (1) Dynamin polymerises into a helical polymer in the absence of GTP; (2) the dynamin polymer constricts in the presence of GTP and (3) dynamin sever membrane upon GTP hydrolysis [52]. Various models for dynamin-mediated membrane fission have been proposed such as the “constrictase model” in which the dynamin helical polymer constricts and mechanically severs the membrane and the “two-stage model” in which constriction and dissociation of dynamin helical polymer are required for membrane cleavage [52]. By using HS-AFM, we and other groups observed cluster formation by dynamin helices upon GTP hydrolysis [71,72]. We also observed that membrane fission occurs between the clustered dynamin helices proposing a novel “clusterase model” [72]. GTP hydrolysis also causes the twisting motion of the dynamin helical polymer that provides torsion at the neck of the endocytic pits to promote membrane fission [73,74]. Thus, dynamin severs membrane by a combination of various mechanical stresses caused by structural changes and depolymerisation upon GTP hydrolysis.

### 4.2. Dynamin 2 Functions in T-Tubule Biogenesis

Dynamin 2 is ubiquitously expressed in various tissues, but its expression level is relatively high in skeletal muscles [75]. In skeletal muscles, dynamin 2 localises to T-tubules at the early stages of development and regulates T-tubule orientation [34,76]. *In cellulo* reconstitution assay for T-tubule-like structures (TLS) revealed that dynamin 2 is required for stabilisation of the TLS [47]. GTPase activity of dynamin 2 is inhibited by BIN1 in a stoichiometry-dependent manner to allow dynamin 2 to stabilise TLS (Figure 1) [34,47]. CNM-associated BIN1 mutants with partially truncated SH3 domain fail to bind to dynamin 2 and induce TLS formation [47]. The expression level of BIN1 is increased as skeletal muscle development progresses, while that of dynamin 2 remains unchanged [19]. Thus, it is interesting to speculate that BIN1 contributes not only to membrane tubulation *per se* but also supports dynamin 2-mediated membrane stabilisation by suppressing GTPase activity to organise the T-tubule system during the normal development of skeletal muscles.

### 4.3. Dysregulation of T-Tubule Function by CNM-Associated Dynamin 2 Variants

*DNM2* is a causal gene for autosomal-dominant CNM and at least 29 pathogenic variants have been identified in the middle, PH, and GE domains [47,77,78,79,80] (Figure 2). Based on the crystal structure of dynamin 1, most of these mutations appear to locate at the interface between the PH domain and the stalk region [63]. As already mentioned in this review, the self-assembly and lipid-binding ability of dynamin are required for efficient membrane fission [81,82,83]. CNM-associated dynamin 2 variants causing mutations in the middle or PH domains formed abnormally stable polymer with elevated lipid binding affinity [47,84,85]. These mutants are gain-of-function because they are featured by elevated GTPase and membrane fission activities [47,86,87,88] (Figure 3). Furthermore, the CNM-associated dynamin 2 mutants induce fragmented T-tubule-like structures in cultured cells because they are resistant to the BIN1-mediated inhibition of GTPase activity [47,88]. Consistently, CNM-model animals (mouse, zebrafish, and fruit fly) expressing mutant dynamin 2 in their skeletal muscles exhibit fragmented or collapsed T-tubules [76,84,85,88,89]. These model animals show reduced calcium release and motor dysfunction that mimic CNM symptoms [84,89]. The molecular dynamics simulation predicts that CNM-associated dynamin 2 mutants form tighter helical structures compared to those with wild type dynamin 2 [90], which may underlie elevated membrane fission activities of CNM-associated dynamin 2 mutants. Further analyses on alterations in structures and dynamics of CNM-associated dynamin 2 mutants will reveal the molecular pathogenesis of CNM.

### 4.4. Correlation between Membrane Fission Activity and Symptom Severities by CNM-Associated Dynamin 2 Variants

*DNM2*-associated CNM represents a wide spectrum of clinical features ranging from severe neonatal forms to moderate adult-onset ones with various histopathological phenotypes [78]. CNM-associated *DNM2* variants are clustered in exons 8, 11, 14 and 16 and the genotype of these variants are potentially correlated with clinical severities [78]. Most reported CNM-associated *DNM2* variants are linked to either early onset and severe phenotype (e.g., p.E368K, p.R369Q and p.S619L) or early onset but milder phenotype (e.g., p.R465W) [78]. In contrast, only a few patients have been reported to develop the late-onset disease. The fission activities of dynamin have been mainly measured based on its GTPase activity and most of the CNM-associated dynamin 2 mutants have been identified as gain-of-function mutants. Interestingly, our quantitative analyses on T-tubule like structures reconstituted *in cellulo* showed a good correlation between membrane fission activities of CNM-associated variants and pathogenicity [91]. Thus, our approach using simple in vitro and *in cellulo* assays together with genetic and clinicopathological analyses should contribute to a more precise diagnosis of pathogenicity, especially when muscle biopsy samples are unavailable (Figure 4). Furthermore, from the therapeutic point of view, early diagnosis by our simple assay may also improve the management and care of these patients.

### 4.5. Other Functions of Dynamin 2 in Skeletal Muscle

In skeletal muscle cells, dynamin 2 functions not only in T-tubule stabilisation but also regulates multiple processes such as vesicle trafficking, cytoskeletal organisation and satellite cell regeneration (Figure 5).

Dynamin 2 regulates clathrin-dependent and -independent endocytosis of glucose transporter-4 (GLUT4) [92], which is required for glucose homeostasis via insulin signalling [93]. In the clathrin-dependent endocytosis, GLUT4 binds to adaptor protein AP2 that recruits clathrin at the plasma membrane, and the clathrin-coated bulk is pinched-off by dynamin 2 [94]. A study using L6 myoblasts demonstrated that dynamin 2 is required for cholesterol-dependent GLUT4 endocytosis [92].

Dynamin 2 is also required for the release of autophagosomes from recycling endosomes and autolysosomes [95,96]. Endocytosed vesicles are normally cleaved by dynamin 2 from early endosomes and transported to the plasma membrane via recycling endosomes [97]. In a starvation condition, recycling endosomes serve as a platform for the assembly of core autophagy-related proteins to induce autophagosome formation [98]. Dynamin 2 directly interacts with LC3, a mammalian ortholog of yeast Atg8, that specifically binds to the autophagosomal membrane via its PH domain [95]. Autophagosomes formed on recycling endosomes are released by dynamin 2 and processed for maturation [95]. In homozygous knock-in mice with a CNM-associated mutant dynamin 2 (R465W), the autophagosome maturation process is defected [99]. Dynamin 2 R465W can still interact with LC3, but its function on autophagosome is impaired, because of enhanced interaction with ITSN1, a binding partner of dynamin 2 on the plasma membrane [95].

In the course of autophagy, dynamin 2 localises not only to recycling endosomes but also localises to autolysosomes [96]. At autolysosomes, the fission activities of dynamin 2 contribute to lipophagy, which is the autophagic degradation of lipid droplet (LD) required for lipid homeostasis [100]. Dynamin 2 depletion or loss of its GTPase activities in hepatocytes results in defective lipophagy [96]. Similarly, loss of dynamin 2 in skeletal muscles also causes defects in lipid homeostasis by altering LD biogenesis and mitochondrial morphology [101]. Dynamin 2 has been implicated in mitochondrial fission cooperatively with Drp1 (dynamin-related protein 1) in COS-7, Sk-Mel2 and HeLa cells [102]. However, dysfunction of CNM-associated dynamin 2 variants in LD biogenesis, lipophagy or mitochondrial fission and their implications in CNM pathogenesis remains to be elucidated.

Dynamin 2 is also implicated in cytoskeletal regulation, especially in the organisation of actin. Dynamin 2 regulates intracellular trafficking of the GLUT4-containing vesicles by controlling actin polymerisation [93]. The actin regulation by dynamin 2 is also required for insulin-dependent exocytosis of GLUT4 to supply intracellular membrane components to T-tubules [103,104,105]. Expression of CNM-associated mutant dynamin 2 disrupts *de novo* actin filament formation in muscle cells [93]. Consistently, in the CNM model mouse expressing CNM mutant dynamin 2 (R465W), translocation of GLUT4 to the plasma membrane is impaired due to disorganised actin filaments, and abnormal perinuclear accumulation of GLUT4 is observed in CNM patient’s muscle biopsy [93].

Actin regulation by dynamin 2 is also required for skeletal muscle development in myoblast fusion [64,66] and the formation of neuromuscular junctions (NMJ) [106]. Invadosomes are actin-rich membrane protrusions required for degradation of the extracellular matrix (ECM), and they play essential roles in myoblast fusion and NMJ formation [107]. In invadosomes, dynamin 2 is involved in actin organisation either by itself via the PR domain [64] or with its interacting proteins such as Tks5 (tyrosine kinase substrate with 5 SH3 domain) [66,106]. Dynamin 2 is also required for the formation and function of invadosomes cooperatively with various BAR domain proteins such as BIN1 [108], endophilin [109] and pacsin 2 [110]. Expression of CNM-associated dynamin 2 mutant (A618T) in C2C12 cells enhances formation of invadosomes with abnormal matrix degradation by inducing F-actin bundles [106].

Costameres, sub-sarcolemmal adhesion sites associated with Z-lines in skeletal muscle, play mechanical and signalling roles during muscle contraction [111]. Costameres consist of multiple components such as integrin [112], actin [113], clathrin [114] and dynamin 2 [115] and they are required for the stabilisation of skeletal muscle fibres by attaching sarcolemma to myofibrils [111]. Dynamin 2 regulates clathrin plaque formation in costameres by interacting with desmin and N-WASP [114,115]. In the CNM-model mouse expressing dynamin 2 mutant and the CNM patient’s biopsy, costameres are defected because of disorganised desmin filaments and clathrin plaques [114,116].

The nuclear positioning to the periphery of skeletal muscle cells requires crosslinking of myofibrils by desmin which is regulated by the arp2/3 complex [117]. Dynamin 2 is required for peripheral nuclear positioning by interacting with N-WASP, an activator of the Arp2/3 complex [51,118,119,120]. CNM mutant dynamin 2 localises around centralised nuclei and their size and numbers are impaired in the adult skeletal muscles in *Dnm2*-KI mice [121,122]. These abnormal nuclei are possibly produced by defective regeneration of satellite cells due to decreased transcription [123]. However, it is still unclear how the function of dynamin 2 around the nuclei is impaired. Further analyses are required for unveiling yet unknown transcriptional regulation by dynamin 2.

### 4.6. Therapeutic Approaches for CNM

CNM-associated dynamin 2 variants cause gain-of-function features in membrane fission activities because of elevated GTPase activity [47,86,87,88]. Likewise, overexpression of wild-type dynamin 2 also induces CNM phenotypes such as muscle weakness, abnormal histology and altered T-tubule structures in mice and *Drosophila* [79,89,119]. Based on these findings, gene silencing approaches are developed to reduce or normalise the expression level of dynamin 2 using AAV-mediated expression of shRNA targeting Dnm2 mRNA or antisense oligonucleotides against Dnm2 pre-mRNA and mRNA [124,125,126]. These gene silencing approaches improve CNM phenotypes of moderate *Dnm2*^R465W/+^ and severe *Dnm2*^S619L/+^ mouse models [124,125,126]. The expression level of dynamin 2 protein is increased in muscle lysates from *Mtm1*-KO mouse and XLMTM1 patients [127]. Therefore, gene silencing approaches targeting *Dnm2* also improved the CNM symptoms in *Mtm1*-KO mice [127,128]. As already mentioned in this review, BIN1 negatively regulates GTPase activities of dynamin 2 in a stoichiometry dependent manner [34,47]. Skeletal muscle-specific *Bin1*-KO mouse shows CNM phenotypes including reduced muscle mass and force, and T-tubule abnormalities with a slight increase of dynamin 2 protein level [34,129]. Thus, downregulation of dynamin 2 by gene silencing tunes its relative amount for BIN1 protein resulting in normal survival, muscular force and triad structures [34,129]. In zebrafish, knockout of a CNM causal gene *SPEG* (striated preferentially expressed protein kinase) that encodes a myosin light chain kinase family protein show T-tubule abnormalities with the increased expression level of dynamin 2 protein [130]. Since SPEG has been shown to interact with MTM1 [5], SPEG may regulate dynamin 2 function together with MTM1 and BIN1 in skeletal muscle. Although it is still unclear if SPEG is also a negative regulator of dynamin 2, gene silencing of *DNM2* may be a potential therapeutic approach for CNM caused by variants in *DNM2* gene as well as for CNM associated with variants in other genes such as *MTM1*, *BIN1*, *SPEG*. Indeed, a clinical trial using investigational antisense medicine DYN101 is ongoing for *DNM2*-associated CNM (NCT04033159).

## 5. Perspectives

In this review, we overviewed the function of BIN1 and dynamin 2 in T-tubule biogenesis and discussed possible molecular mechanisms of CNM pathogenesis caused by their membrane remodelling defects. Abnormal membrane remodelling by CNM-associated variants of BIN1 and dynamin 2 has been greatly elucidated using multidisciplinary approaches. However, the impact of CNM-associated variants on multifunctional features of dynamin 2 at various cellular organelles is still largely unknown. A comprehensive understanding of dysregulated functions of dynamin 2 in the multiple cellular processes may contribute to a better elucidation of pathomechanisms of CNM and the development of more precise diagnosis, management and care of CNM patients. Although we focused on the T-tubule biogenesis by BIN1 and dynamin 2, there are a variety of other proteins involved in T-tubule formation, and many of them are associated with muscle diseases [6,131]. A more comprehensive understanding of protein functions that affect T-tubule formation is required for a better understanding of the CNM pathogenesis caused by abnormal membrane remodelling.

## Figures and Tables

**Figure 1 ijms-23-06274-f001:**
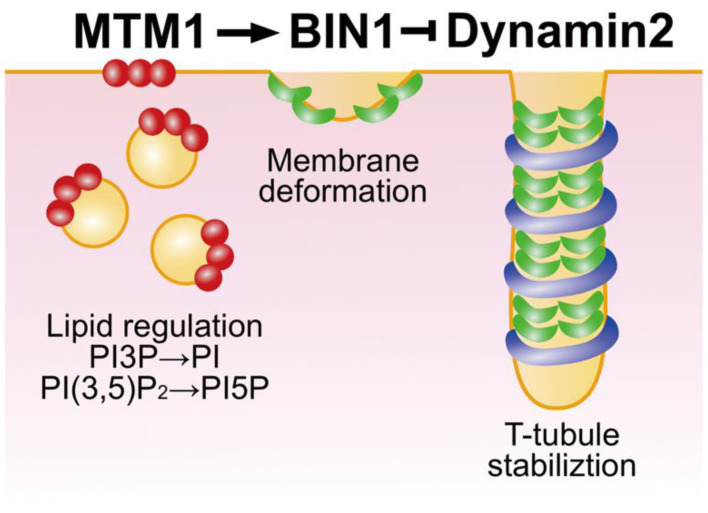
Functions of MTM1, BIN1 and DNM2 in T-tubule biogenesis. CNM causative genes MTM1, BIN1 and DNM2 contribute to T-tubule biogenesis in a common pathway by respectively regulating lipid homeostasis, membrane deformation and T-tubule stabilisation.

**Figure 2 ijms-23-06274-f002:**
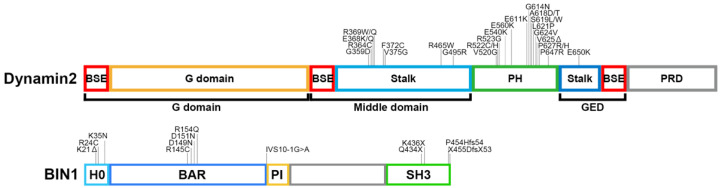
Domain structures of dynamin 2 and BIN1. Schematic illustrations of domain structures and CNM-associated mutations in dynamin 2 and BIN1.

**Figure 3 ijms-23-06274-f003:**
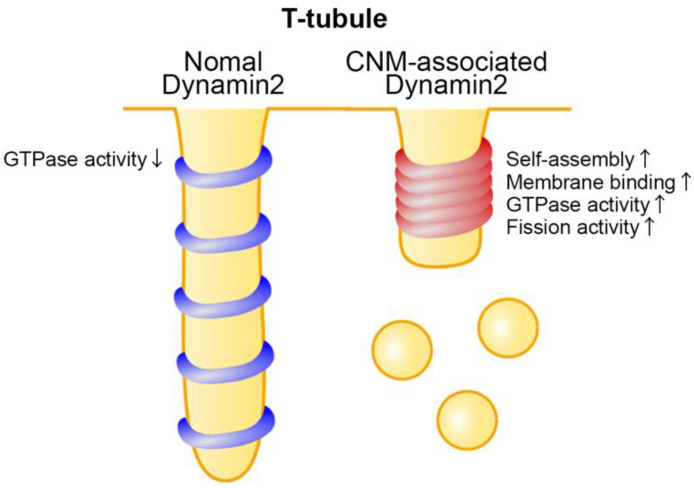
Possible mechanisms of defective T-tubule formation caused by CNM-associated dynamin 2 mutant. CNM-associated dynamin 2 exhibits gain-of-function features with elevated GTPase and membrane fission activities compared to normal dynamin 2.

**Figure 4 ijms-23-06274-f004:**
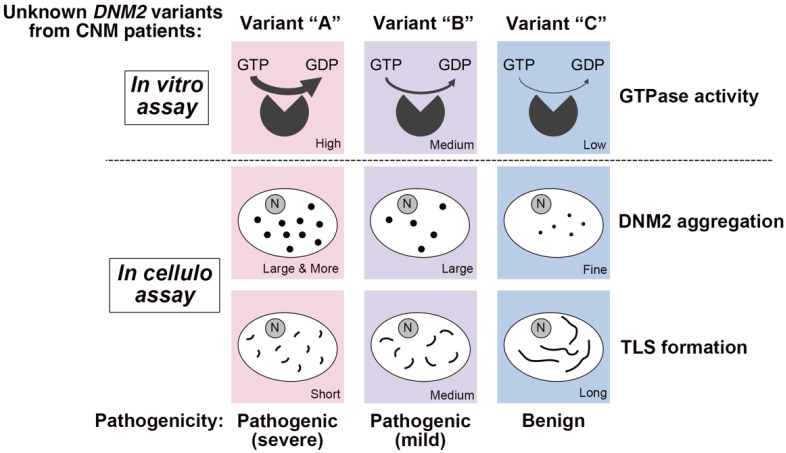
Determining pathogenicity of novel CNM variants by various analyses. Possible phenotypic summary of unknown variants identified from CNM patients analysed by various assays either in vitro (GTPase activity) or *in cellulo* (DNM2 aggregation and TLS formation) to determine their pathogenicity. N: nuclei.

**Figure 5 ijms-23-06274-f005:**
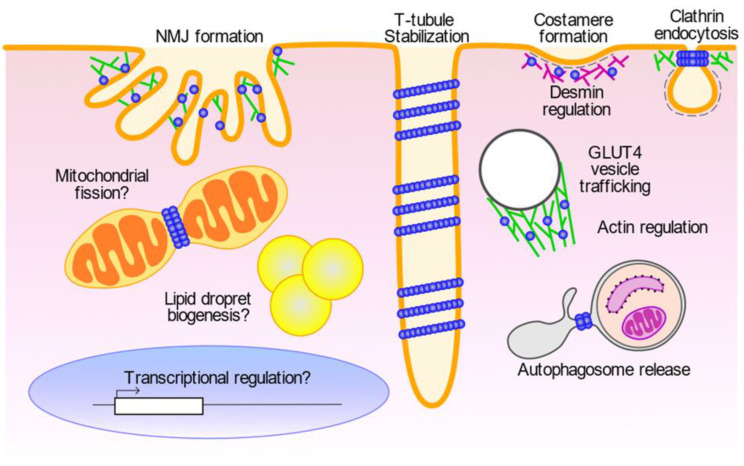
Multiple functions of dynamin 2 in skeletal muscle cells. Dynamin 2 (blue) is involved in multiple processes in muscle cells such as T-tubule biogenesis, NMJ formation, costamere formation, endocytosis and vesicle trafficking, autophagy and lipid homeostasis.

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
