# Peer review of "Centronuclear Myopathy Caused by Defective Membrane Remodelling of Dynamin 2 and BIN1 Variants"

_ijms, 2022, doi:10.3390/ijms23116274_

Round 1
Reviewer 1 Report
Fujise et al., in their review titled: "Centronuclear myopathy caused by defective membrane remodelling by dynamin 2 and BIN1 variants", described well the mechanisms in volved in different stages of Centronuclear myopathy focusing above all on the role of dynamin 2 and BIN1 variants. I appreciate very much the discussion on the role of the single gene variants. I have no major comments but I suggest only a fast revision of some English phrases that seem to be too long.Author Response
We thank Reviewer 1 for the very supportive comments. We have amended some English phrases to shorter.
Reviewer 2 Report
The authors introduced the functions of DNM2 and BIN1 in T-tubule biogenesis and mainly discussed the potential mechanisms associated with DNM2 and BIN1 membrane remodeling in the pathogenesis of centronuclear myopathy (CNM). However, this review adds little to our current understanding of the defective membrane remodeling caused by dynamin 2 and BIN1, particularly when compared to previously published reviews such as Int J Mol Sci. 2021 Oct 21;22(21):11377; Nat Rev Mol Cell Biol. 2021 Nov;22(11):713-732; Neurotherapeutics. 2018 Oct;15(4):966-975; Front Aging Neurosci. 2014 Dec 19;6:339; PLoS Genet. 2012;8(4):e1002595; J Mol Med (Berl). 2010 Apr;88(4):339-50. Thus, I do not feel that the manuscript is appropriate for publication in the International Journal of Molecular Sciences.
Author Response
We thank Reviewer 2 for the honest opinion about our review. We very much appreciate the previously published excellent reviews on CNM including the ones listed by the reviewer. In our review article, we aimed to compensate for the content that is not covered by other reviews. However, because of the nature of the review article, it is very difficult to avoid excluding redundant information. To make our standpoint clearer, all the reviews listed above were cited and our statement was added at the end of the introduction section.
Round 2
Reviewer 2 Report
The authors introduced the functions of DNM2 and BIN1 in T-tubule biogenesis and mainly discussed the potential mechanisms associated with DNM2 and BIN1 membrane remodeling in the pathogenesis of centronuclear myopathy (CNM). The previous version is simple and fragmented, and the revised version does not provide any substantial improvements. Thus, I strongly feel that the manuscript is not appropriate for publication in the International Journal of Molecular Sciences.
Major points:
- T-tubules part 2. Both DNM2 and BIN1 are involved with T-tubule biogenesis. The author should carefully introduce the process of T-tubule biogenesis, and the roles of both proteins in this process. Other proteins (including CAV3, DYSF, MG29, JPH1, MTM1) are also part of this T-tubule biogenesis process. RyR1 localizes in the SR, and it contacts with DHPR at T tubule to form the triad. Images demonstrating how to form T-tubules, how to connect SR and form triads, and how to conduct the function of E-C coupling are missing. The author should give a brief introduction of this process and focus on DNM2 and BIN1. However, the author directly jumps from sarcolemmal invaginations to E-C coupling.
- Therapeutic part at 4.6. The authors mentioned MTM1, BIN1, DNM2, and SPEG mice, and DNM2 reduction using AAV-mediated expression of shRNA targeting Dnm2 mRNA or antisense oligonucleotides against Dnm2 could improve the CNM phenotypes. What are the potential mechanisms behind them? Is the mechanism the same for all four proteins, or are there differences? The description is overly simplistic and devoid of insight.
Overall, this review adds little to our current understanding of the defective membrane remodeling caused by dynamin 2 and BIN1, particularly when compared to previously published reviews such as Int J Mol Sci. 2021 Oct 21;22(21):11377; Nat Rev Mol Cell Biol. 2021 Nov;22(11):713-732; Neurotherapeutics. 2018 Oct;15(4):966-975; Front Aging Neurosci. 2014 Dec 19;6:339; PLoS Genet. 2012;8(4):e1002595; J Mol Med (Berl). 2010 Apr;88(4):339-50.
Author Response
*This round of review is based on the Academic Editor's comments and our response to reviewer 2 remains the same as the one we used in the 1st round of review (Please see below).
We thank Reviewer 2 for the honest opinion about our review. We very much appreciate the previously published excellent reviews on CNM including the ones listed by the reviewer. In our review article, we aimed to compensate for the content that is not covered by other reviews. However, because of the nature of the review article, it is very difficult to avoid excluding redundant information. To make our standpoint clearer, all the reviews listed above were cited and our statement was added at the end of the introduction section.